# The Effect of Hyperbaric Storage on the Nutritional Value and Retention of Certain Bioactive Proteins in Human Milk

**DOI:** 10.3390/nu16101455

**Published:** 2024-05-12

**Authors:** Katarzyna Mazur, Barbara Kusznierewicz, Dorota Martysiak-Żurowska, Izabela Drążkowska, Edyta Malinowska-Pańczyk

**Affiliations:** 1Department of Food Chemistry, Technology and Biotechnology, Chemical Faculty, Gdansk University of Technology, G. Narutowicza 11/12 Str., 80-233 Gdańsk, Poland; katarzyna.mazur1@pg.edu.pl (K.M.); barbara.kusznierewicz@pg.edu.pl (B.K.); dorota.martysiak-zurowska@pg.edu.pl (D.M.-Ż.); 2Division of Neonatology, Medical University of Gdańsk, 80-210 Gdańsk, Poland; izabela.zapasnik@gumed.edu.pl

**Keywords:** human milk, hyperbaric storage at subzero temperature, proteins, carbohydrates, fat, energy value, lactoferrin, lysozyme, *α-*lactalbumin, secretory immunoglobulin A

## Abstract

Human milk (HM) contains the essential macronutrients and bioactive compounds necessary for the normal growth and development of newborns. The milk collected by human milk banks is stored frozen and pasteurized, reducing its nutritional and biological value. The purpose of this study was to determine the effect of hyperbaric storage at subzero temperatures (HS-ST) on the macronutrients and bioactive proteins in HM. As control samples, HM was stored at the same temperatures under 0.1 MPa. A Miris HM analyzer was used to determine the macronutrients and the energy value. The lactoferrin (LF), lysozyme (LYZ) and *α-*lactalbumin (*α-*LAC) content was checked using high-performance liquid chromatography, and an ELISA test was used to quantify secretory immunoglobulin A (sIgA). The results showed that the macronutrient content did not change significantly after 90 days of storage at 60 MPa/−5 °C, 78 MPa/−7 °C, 111 MPa/−10 °C or 130 MPa/−12 °C. Retention higher than 90% of LYZ, *α-*LAC, LF and sIgA was observed in the HM stored at conditions of up to 111 MPa/−10 °C. However, at 130 MPa/−12 °C, there was a reduction in LYZ and LF, by 39 and 89%, respectively. The storage of HM at subzero temperatures at 0.1 MPa did not affect the content of carbohydrates or crude and true protein. For fat and the energy value, significant decreases were observed at −5 °C after 90 days of storage.

## 1. Introduction

Human milk (HM) is a dynamic biofluid which contains all the essential nutrients and is an excellent source of bioactive substances [1]. In particular, the presence of the latter during breastfeeding strengthens the immature immune systems of newborns and protects them against respiratory and gastrointestinal infections [2]. Due to several benefits of feeding infants with HM, it is important to ensure access to this food, even when mothers cannot feed their babies directly. In this case, nursing with milk from human milk banks (HMBs) is the best option. The procedure currently used by HMBs to preserve HM involves many steps that negatively affect its nutritional value and bioactive components. Storing milk at −20 °C for the first three months after its lactation causes the loss of certain biologically active components like lactoferrin (LF), lysozyme (LYZ), immunoglobulin A (IgA) and other nutrients such as fats or carbohydrates [3,4,5]. However, the step that has the greatest impact on the deterioration of HM, but ensures its microbiological safety, is Holder pasteurization (HoP) at 62.5 °C for 30 min [6]. Therefore, numerous research groups around the world are looking for other methods for preserving HM that will not cause the loss of its important components with biological properties.

One proposed solution is to replace pasteurization with high-pressure processing (HPP). The treatment of HM at 500–600 MPa for 8 min effectively inactivates the HM microbiota, but the retention of its bioactive components is at a similar level to that after HoP [7]. It has also been shown that the content of bioactive components in HM does not change after up to 30 min of pressure at up to 300 MPa. However, such short-term exposure to moderate pressure does not ensure microbiological safety. A novel approach to achieving the expected level of inactivation at moderate pressure may be to significantly extend the holding time under these conditions. This concept is known as hyperbaric storage (HS). There are now a growing number of reports on the potential use of this new technique in the food industry as a method of extending the shelf life of many food products [8]. In our previous work, we showed that the storage of HM at moderate pressures of up to 130 MPa at subzero temperatures led to the inactivation of its microbiota to the level required by the European Milk Bank Association [9]. Therefore, the use of this new technique may make it possible to remove the two steps of freezing/thawing and thermal pasteurization from the milk handling procedures at HMBs and replace them with one-step storage under moderate pressure. However, before implementing this method, it is necessary to determine what changes occur in the milk’s components during storage under such specific conditions.

The aim of this study was to determine the changes in the content of proteins, carbohydrates, fats and the energy value, as well as the concentrations of selected bioactive proteins, LF, LYZ, *α-*lactalbumin (*α-*LAC) and sIgA, in HM stored for 90 days under moderate pressure at subzero temperatures, in conditions where water remains unfrozen [10].

## 2. Materials and Methods

### 2.1. Materials

Frozen HM was obtained from voluntary milk donors, who gave birth at the University Clinical Hospital in Gdańsk. The characteristics of the donors are shown in Table 1. When interviewing the donors, attention was paid to the age of the women, the length of the pregnancy and the stage of lactation. All the experimental procedures were approved by the Rector’s Commission for Ethics in Research with Human Participation at the Gdańsk University of Technology 20 January 2022. The patients provided written consent to participate in the study.

### 2.2. Sample Preparation for Storage

Using an electric breast pump, the milk used for the study was lactated by the donors into sterile containers designed for storing HM, maintaining hygienic standards, and immediately frozen at −20 °C. The HM was then delivered to the laboratory while maintaining the cold chain (the transport time did not exceed 1 h) and stored at −20 °C for a maximum of 4 weeks. Then, randomly selected HM samples were thawed at 5 °C for 24 h and pooled into four large batches of 3 L. Each batch of HM was then bottled into 25 flexible milk storage bottles (80 mL capacity) (Medela 200.2957) (Figure 1) and kept in an ice bath until placed in the pressure vessels or in the freezer (no longer than 1 h). For HS, the bottles had to be filled to the maximum so that no free space remained inside. For the samples stored at subzero temperatures under atmospheric pressure, the bottles were filled to a capacity of 80 mL so that no damage to the container would occur after freezing.

### 2.3. Storage

Pressure was generated as described in Patent Application Number P.447629 [11]. The milk samples were stored in the equipment designed at the Department of Food Chemistry, Technology and Biotechnology, the Gdańsk University of Technology, and built by DS Technology Ltd. (Słupsk, Poland). Milk from the two different batches was placed into 620 mL pressure vessels that were filled with distilled water and then secured with a spring that kept the samples in the upper unfrozen zone and sealed tightly. The pressure vessels were placed from the side of the closure in a cryogenic bath and gradually immersed to cool the system and generate pressure. The magnitude of the pressure generated in such a system depends on the temperature [10]. The total time taken for this step was 90 min. After this, the pressure vessels were transferred into a storage section and stored for 7, 14, 21, 28, 63 and 90 days. Six pressure vessels were prepared for each condition: 60 MPa/−5 °C, 78 MPa/−7 °C, 111 MPa/−10 °C and 130 MPa/−12 °C. In parallel, samples of the same milk were stored at the same temperatures under atmospheric pressure (Figure 2). After the specified time, the pressure vessels were removed from the storage zone and placed in a water bath at 20 °C for 60 min. The milk samples were taken from the pressure vessels and stored in an ice bath prior to determining their protein, carbohydrate and fat content and energy value, as well as their bioactive protein concentrations. Samples stored at subzero temperatures under atmospheric pressure were also thawed in a water bath at 20 °C for 60 min.

### 2.4. Determination of Essential Macronutrients and the Energy Value

For the analysis of the content of macronutrients in the control and stored HM and its energy value, a Miris human milk analyzer (HMA) (Miris AB, Uppsala, Sweden), based on semi-solid-state mid-infrared (MIR) transmission spectroscopy, was used. The amounts of fat (g/100 mL), crude protein (g/100 mL), true protein (g/100 mL) and carbohydrates (g/100 mL) and the energy value (kcal/100 mL) were determined. According to the Miris HMA user manual, crude protein, also referred to as total protein, is the protein content based on the total amount of nitrogen (N) in a sample, while true protein represents only the content of actual protein (based only on protein N). Each sample was heated to 40 °C in a thermostatic bath before analysis and then homogenized using a Miris sonicator. The repeatability of the instrument was determined based on the information published in the Miris HMA manual: fat, protein ≤ 0.05 g/100 mL; carbohydrate ≤ 0.08 g/100 mL. The precision of the protein, fat and carbohydrate determination using the Miris HMA was 10.51%, 8.31% and 3.49%, respectively [12]. All the determinations were carried out twice in duplicate.

### 2.5. Determination of α-LAC, LYZ and LF Content Using High-Performance Liquid Chromatography (HPLC)

The frozen and HS-ST milk samples were centrifuged (4 °C, 12,000× *g*, 15 min), and clear supernatant was collected and transferred into vials. The samples were analyzed using HPLC in an Agilent Technologies 1200 Series chromatograph with a DAD detector. Separation was performed on a Kinetex XB-C18 chromatographic column (4.6 × 150 mm, 5 µm, 100A, Phenomenex, Torrance, CA, USA) with a guard column SB-C18 (4.6 × 12 mm, Phenomenex, USA). The analysis was carried out in linear gradient elution mode from 70% A:30% B (0 min) to 50% A:50% B (20 min) and then to 20% A: 80% B (25 min), where eluent A was 0.1% trifluoroacetic acid in distilled water and eluent B was a 0.1% solution of trifluoroacetic acid in acetonitrile. The volume flow was 1 mL/min. The *α-*LAC, LF and LYZ contents of the milk were calculated using a calibration curve where the peak area was plotted against a certain protein concentration in standard solutions (human lysozyme; lactoferrin from human milk; *α-*lactalbumin from human milk, Sigma-Aldrich, St. Louis, MO, USA). The results were expressed as g/100 mL of the sample. All determinations were carried out twice in triplicate.

### 2.6. Determination of sIgA Content

The concentration of sIgA was determined using an ELISA kit (IDK^®^ sIgA ELISA, Immundiagnostik AG, Bensheim, Germany) according to the protocol provided by the manufacturer. Prior to this, the milk samples were thawed, centrifuged (4 °C, 3000× *g*, 10 min) and then diluted 1:20,000. The data were expressed in g/100 mL of milk. All determinations were carried out twice in triplicate.

### 2.7. Statistical Analysis

The results in the figures and tables are presented as mean values and standard deviation (mean ± SD). To test the statistical significance of the differences observed in the macronutrients and bioactive proteins between the samples stored at different temperatures under atmospheric and moderate pressure, a two-way analysis of variance (ANOVA) was conducted. Differences were considered statistically significant at *p* < 0.05. To determine the differences between pairs of groups, Tukey’s test was performed. The statistical analysis was conducted using Prism version 10.2.0 (341) (GraphPad Software, Inc., San Diego, CA, USA).

## 3. Results and Discussion

### 3.1. Macronutrient Content and Energy Value of HM

The macronutrient content of HM depends on the mother’s diet, her ethnicity and other factors [13]. It has been shown that the average concentration of individual nutrients ranges from 0.85 to 3.60 g/100 mL for proteins, 2.91 to 5.20 g/100 mL for fats and 5.52 to 8.01 g/100 mL for carbohydrates, with an energy value of 60.99–81.4 kcal/100 mL [14,15,16,17,18,19,20]. The average concentrations of these compounds and the energy value in the four batches of tested, pooled HM are shown in Table 2. The values obtained are within the ranges described in the literature determined for individual milk samples.

These components in HM provide proper nutritional value and are a source of the bioactive factors that ensure the normal growth and healthy development of infants. Therefore, it is important that the methods used to ensure the microbiological safety of milk in HMB do not lead to a reduction in macronutrient content. The data on changes in the macronutrients of HM mainly concern fresh milk stored at −20 and −80 °C, milk immediately after HoP and milk stored in a freezer after HoP [21,22]. Limited information can be found on its macronutrient changes after HPP [23,24]. On the other hand, data on how the nutritional value of milk changes during further storage after HPP are, to the best of our knowledge, completely unavailable. This also applies to milk stored in hyperbaric conditions.

#### 3.1.1. Effect of HS-ST on Proteins

In general, the changes in the crude protein content in the HM during HS-ST were not statistically significant (*p* > 0.05). The values oscillated during the measurements at different storage times, and the variations ranged from −5.6% to +3.7% for 60 MPa/−5 °C, −3.8% to +7.7% for 78 MPa/−7 °C, −3.7% to +3.7% for 111 MPa/−10 °C and +7.7% for 130 MPa/−12 °C (Figure 3). For true protein, a similar pattern of changes was observed, although the percentages of the variations were smaller. On the last day of storage, the true protein content of the HM decreased by 4.5% at 60 MPa/−5 °C and 111 MPa/−10 °C, increased by 4.8% at 78 MPa/−7 °C and did not change at 130 MPa/−12 °C (Figure 4). However, these alterations were not statistically significant (*p* > 0.05). The lack of variation in the protein content of milk immediately after pressurization at 500 MPa for 8 min or HoP, respectively, were shown by [25,26], respectively. In contrast, immediately after HoP, [21] and [19] found significant reductions in the protein content of HM of 3.9% and 2.5%, respectively.

The concentration of crude and true protein in the sample of HM stored for 90 days under atmospheric pressure at subzero temperatures from −5 to −12 °C also oscillated over time, but these changes were irrelevant (*p* > 0.05) (Figure 3 and Figure 4). No change in the protein content was also shown after 24 weeks of storage of raw HM at −20 and −80 °C [22]. Similarly, in pasteurized HM stored at −20 °C for 6 months, no change in the protein concentration was noticed [4]. However, a 13.4% increase in the protein content compared to fresh HM was observed in pasteurized HM stored for 8 months at −20 °C [27]. These differences in the results obtained by different authors may be due to various factors, such as differences in the quantitative composition of HM and in the preparation of the samples for testing. According to [4], a possible reason for the high variation in the crude protein content was the nonoptimal homogenization of the milk samples before the measurements. After long periods of frozen storage, casein micelles become destabilized and there are abnormalities in the protein structure, leading to its precipitation, so the sample homogenization time used may not have been sufficient to properly determine the protein content. Explaining the reasons for changes in the protein content of HM will not only allow precise quantification but also indicate changes that may affect the nutritional and biological value of HM proteins after processing or storage.

#### 3.1.2. Effect of HS-ST on Carbohydrates

The carbohydrate content did not change significantly after HS for up to 90 days under all the conditions used, although the observed changes were −1.2% to +0.6% for 60 MPa/−5 °C, −3.9% to −0.3% for 78 MPa/−7 °C, −4.5% to +0.6% for 111 MPa/−10 °C and −5% to −0.6% for 130 MPa/−12 °C. For pressurized HM, [25] showed a significant reduction in the carbohydrate concentration in milk treated at 500 MPa for 8 min. For Holder-pasteurized HM, it was shown that after this process the carbohydrate content did not change significantly [19]. According to [27], long-term storage (up to 12 moths) of pasteurized HM at −20 °C does not lead to significant changes in its carbohydrate content. However, a small but significant reduction in the carbohydrate concentration of HM (by 1.7%) was observed by [4] after 6 months of freezer storage. A greater loss (3.1%) was shown by [28] after 3 months of pasteurized HM storage. In our study, we stored milk at atmospheric pressure at higher subzero temperatures, and we noted that in these conditions a decrease in the concentration of carbohydrates took place. These differences ranged from 0.3% to 8.4% and were significant (*p* < 0.05) for HM stored at −5 °C for 90 days and during storage at −7 °C and −12 °C. Relevant changes were not observed in the milk stored at 0.1 MPa/−10 °C (Figure 5).

#### 3.1.3. Effect of HS-ST on Fat and the Energy Value

The fat content of the hyperbarically stored HM varied according to the conditions used (Figure 6). The changes ranged from −2.4% to +13% on day 90 at 60 MPa/−5 °C and on day 63 at 111 MPa/−10 °C. However, these changes were not statistically significant. A reduction in fat content, ranging from 3.5 to 25%, was observed in the HM after HoP [4,21,26]. Such changes do not occur in high-pressure-treated HM [24,25]. Also, there was a reduction in the milk fat content in both the raw and pasteurized HM stored frozen. Although various papers have used different storage times, from 24 h to 12 months, most have shown that there is a significant reduction in fat of 2.4–11.3% after freezing at −20 °C and −80 °C [4,8,22]. The longer the milk was stored, the greater the losses reported. In our control samples, the HM was stored at higher temperatures from −12 to −5 °C under atmospheric pressure. In these conditions, the fat content also varied from −16.2% to +6.1% (Figure 6). In the milk stored at −5 °C, the reduction was significant (*p* < 0.05) at about 15–16% after 63 and 90 days of storage. At lower temperatures, the changes were not significant (*p* > 0.05). The freezing/thawing processes of HM can alter the structure of fat globules, which lead to coalescence and facilitate its adhesion to the vessel walls [21]. Additionally, damage to the membrane of fat globules facilitates the action of lipases, leading to increased concentrations of free fatty acids, which cannot be quantified using the Miris HMA. Ref. [29] showed that lipoprotein lipase and bile-salt-stimulated lipase remain active after the storage of raw milk at −20 °C for 5 months and lead to hydrolysis of the milk fat. Presumably, the higher the temperature, the higher the activity of these enzymes will be [30]; hence, the greatest changes are observed at the highest temperature used. It is interesting that HS-ST of the HM did not change its fat content. This may be due to the fact that milk under these conditions remains in a liquid state, and these conditions prevent coalescence or the activity of lipases is inhibited. Further work is needed to explain this phenomenon.

About 50% of the energy value of HM originates from fat [26]. Therefore, changes in the energy value of HM (Figure 7) were correlated with fat loss, especially in the milk samples where there was a significant reduction in this component. As with the fat determinations, statistically significant changes in the energy value of the milk were recorded in the samples stored at 0.1 MPa/−5 °C. Pearson’s correlation coefficient for these samples was greater than 0.9 (*p* < 0.05).

### 3.2. Bioactive Protein Content of HM

The bioactive protein content of HM varies with lactation time [31]. In mature HM, the concentrations of *α-*LAC, LF, LYZ and sIgA are about 0.2–0.48 g/100 mL [32,33], 0.2–0.26 g/100 mL [3], 0.015–0.024 g/100 mL [34] and 0.07–0.149 g/100 mL, respectively [35]. The concentrations of these proteins in the tested HM are shown in Table 2 and are consistent with those reported in the literature.

#### 3.2.1. Effect of HS-ST on *α-*LAC

*α-*LAC plays an important role in proper infant development. This protein provides essential amino acids, primarily tryptophan, which influence the synthesis of the serotonin and melatonin neurotransmitters, enabling the regulation of infants’ sleep cycle. In addition, cysteine, derived from the degradation of *α-*LAC, is a precursor for glutathione, which exhibits anti-inflammatory and antioxidant properties [33]. It can be noted that the storage of HM under HT-ST did not cause significant differences in the content of *α-*LAC throughout the storage period (*p* > 0.05), especially for 60 MPa/−5 °C, 78 MPa/−7 °C and 130 MPa/−12 °C. For HM stored at 111 MPa/−10 °C, a slight but not significant (*p* > 0.05) increase in its content was observed at a level of about 8%. Storage of HM at the same temperatures but under atmospheric pressure did not change the concentration of *α-*LAC, except in HM kept at 0.1 MPa/−5 °C. As shown in Figure 8, after 21 days of HM storage, the content of *α-*LAC reduced by 17% and did not change significantly by day 90. There are no data in the available literature on changes in *α-*LAC in stored frozen HM or immediately after pressurization. In our earlier studies, we showed that HoP did not cause significant changes in the *α-*LAC concentration [36]. For bovine milk, it was found that there was an 8% decrease in *α-*LAC after storage at −20 °C for 30 days [37]. This protein in bovine milk was also resistant to high pressure up to 700 MPa [38].

#### 3.2.2. Effect of HS-ST on LF

LF is a protein that is resistant to digestion in the gastrointestinal tract and exhibits immunomodulatory, antiviral and antibacterial activity. The latter property of LF is directly related to the ability of this protein to bind iron, which inhibits the proliferation of pathogens that require iron for growth. In addition, LF can act as a transcription factor for genes encoding certain cytokines, thereby modulating the function of infants’ immune systems [31]. The concentration of LF in HM stored hyperbarically did not change after 90 days at 60 MPa/−5 °C, 78 MPa/−7 °C or 111 MPa/−10 °C (*p* < 0.05). A significant reduction in this protein occurred when 130 MPa/−12 °C was applied. In these conditions, the LF concentration had already decreased by 65% after 21 days of storage, and after 90 days, the retention was only 14% (Figure 9). Probably, in these conditions, the denaturation of LF took place. A significant reduction in LF content, by 37%, after the pressurization of HM at 425 MPa (4 cycles of 6 min) was obtained in [6] when the process was conducted at 4 °C. Pressurization of HM at 37 °C retained 83% of its LF content. Also, [25] obtained 72% retention of this protein in HM treated with 500 MPa for 8 min at 4 °C. After HoP, the average recovery of LF in the HM samples ranged from 15% to 40% [6,39,40,41].

For the HM samples stored at subzero temperatures under atmospheric pressure, significant changes in LF content were observed at −5 °C. After 90 days, the LF content decreased by 33%. The changes in the LF content at lower temperatures (−7 °C to −12 °C) ranged from −28% to +16% but were not relevant compared to the initial content (*p* < 0.05). Ref. [3] observed a 37% and 54% reduction in the LF content in HM after storage at −20 °C for 3 and 6 months, respectively. Even greater reductions in this protein (55% and 75%, respectively) in HM stored at −20 °C were reported by [42]. We assume that moderate pressure up to 111 MPa at subzero temperatures may limit the loss of this protein during long-term storage.

#### 3.2.3. Effect of HS-ST on LYZ

LYZ is a protein that protects infants from pathogens by hydrolyzing the bond between 1-4-N-acetyl-glycosamine and muramic acid in the cell walls of Gram-positive bacteria [43]. It also shows a synergistic effect together with LF, and as the result of the activity of both proteins, the growth of Gram-negative bacteria is also inhibited [44]. The LYZ content in the HM varied during HS. The magnitude and direction of changes depended on the pressure–temperature conditions (Figure 10). Pressure of 60 MPa/−5 °C resulted in a significant increase in the LYZ content by 63 days of storage, and a decrease in LYZ was observed thereafter, although this change was insignificant (*p* > 0.05). However, at day 90, the LYZ concentration was 37% higher than in the initial HM. At a higher pressure, 78 MPa/−7 °C and 111 MPa/−10 °C, the changes in LYZ were smaller than at 60 MPa/−5 °C. In the first month, a slight increase in the LYZ content was observed in the HM stored in these conditions, and then the concentration decreased to the same level as in the initial HM. A reduction in the LYZ content was found in the HM stored at 130 MPa/−12 °C. After 90 days, the LYZ concentration of these samples was lower by 33%. [6] showed that a high pressure of 425 MPa at 4 °C and 37 °C (4 cycles of 6 min) caused a decrease in the LYZ level by about 6% and 2%, respectively. A greater loss of LYZ in the HM was obtained after HoP, for which only about 50% retention of this protein was found [41].

In the HM stored at subzero temperatures under atmospheric pressure, we noticed a slight reduction in the LYZ content at −5 °C (*p* < 0.05). At lower temperatures, the concentration of this protein did not change (Figure 10).

#### 3.2.4. Effect of HS-ST on sIgA

sIgA plays the first very important role in the immune system, primarily exhibiting anti-inflammatory effects, inhibiting the colonization of pathogens in the gastrointestinal tract of infants and protecting against the penetration of harmful soluble substances [43]. The content of sIgA did not change during storage at subzero temperatures under atmospheric or moderate pressure (Figure 11). Only in the samples stored at 0.1 MPa/−5 °C was a slight reduction in the sIgA content noticeable, but these changes were not relevant. In comparison, HoP causes a decrease in sIgA by 14% to 30%, as reported by various authors [41,45,46]. Differences in the retention of this component after HoP likely depend on several factors, such as the time taken to reach the pasteurization temperature, the cooling time and the volume of HM processed [46].

## 4. Conclusions

In conclusion, our results show that HS-ST of HM in conditions where water does not crystallize allows for preservation of the high nutritional value of HM. The contents of protein, carbohydrates and fat and the energy value of HM do not change during 90 days of storage at pressure up to 130 MPa/−12 °C. In the case of bioactive proteins, significant retention of *α-*LAC, LYZ, LF and sIgA was observed. The smallest changes in these components occur during HS at pressures of up to 111 MPa. At a higher pressure and a lower temperature (130 MPa/−12 °C), there is a significant reduction in LYZ and LF, by 39 and 89%, respectively. In milk stored at the same temperature at atmospheric pressure, in conditions where large ice crystals form, the greatest negative changes occur at −5 °C. Under these conditions, a reduction in the fat content and energy value was observed. Appropriate HS-ST conditions are one of the most promising non-thermal methods for preserving HM. This method not only eliminates its microbiota but also does not change its nutritional and biological value. Before using this method, further studies are needed to evaluate the effects of these conditions on viruses and other bioactive components of HM, such as antioxidant enzymes, cytokines, etc.

## Figures and Tables

**Figure 1 nutrients-16-01455-f001:**
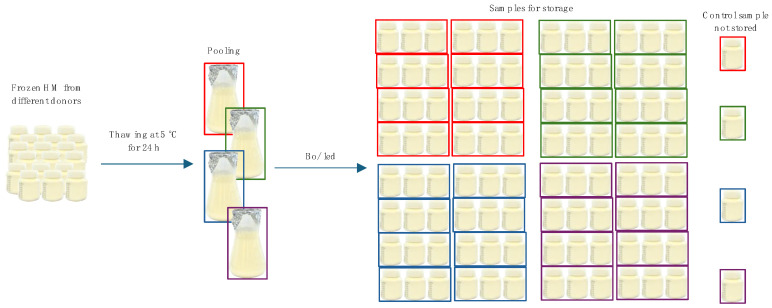
Sample preparation.

**Figure 2 nutrients-16-01455-f002:**
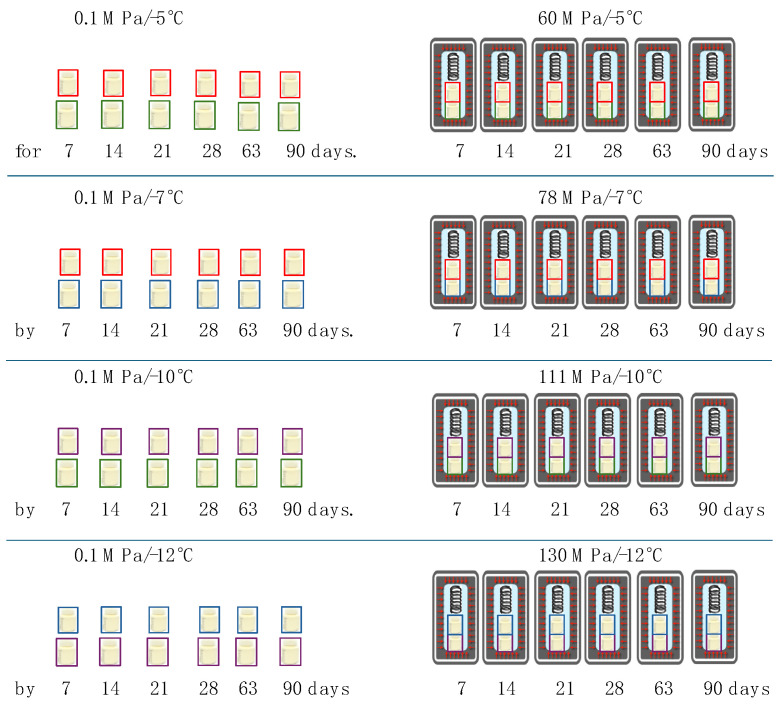
Sample storage scheme.

**Figure 3 nutrients-16-01455-f003:**
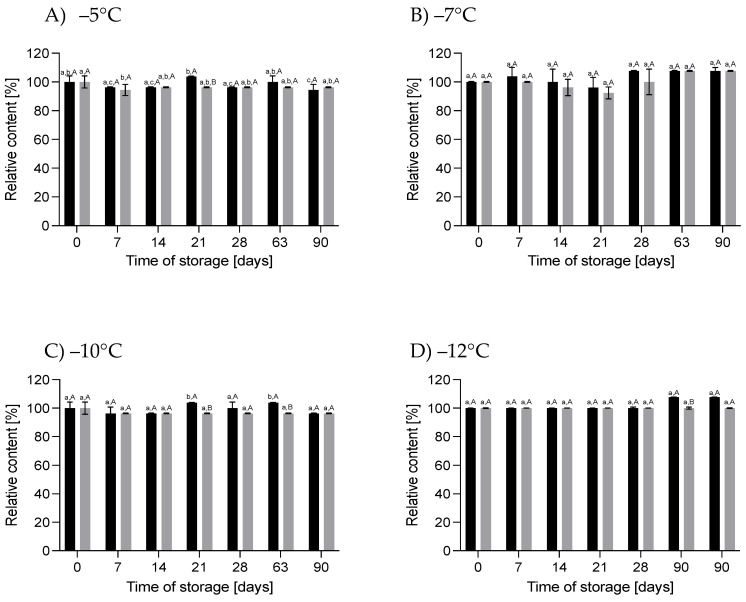
Changes in the relative content of crude protein during storage at subzero temperatures under increased pressure (dark bars on each chart) and at atmospheric pressure (light bars on each chart); different lowercase letters (a–c) indicate statistically significant differences among the different storage times for the same treatment, and capital letters (A,B) indicate statistically significant differences among various treatments at the same time (*p* < 0.05).

**Figure 4 nutrients-16-01455-f004:**
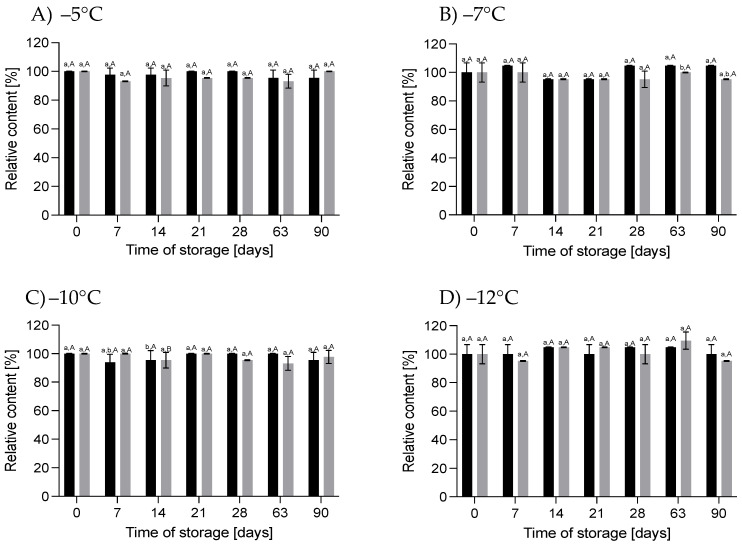
Changes in the relative content of true protein during storage at subzero temperatures under increased pressure (dark bars on each chart) and at atmospheric pressure (light bars on each chart); different lowercase letters (a,b) indicate statistically significant differences among the different storage times for the same treatment, and capital letters (A,B) indicate statistically significant differences among the various treatments at the same time (*p* < 0.05).

**Figure 5 nutrients-16-01455-f005:**
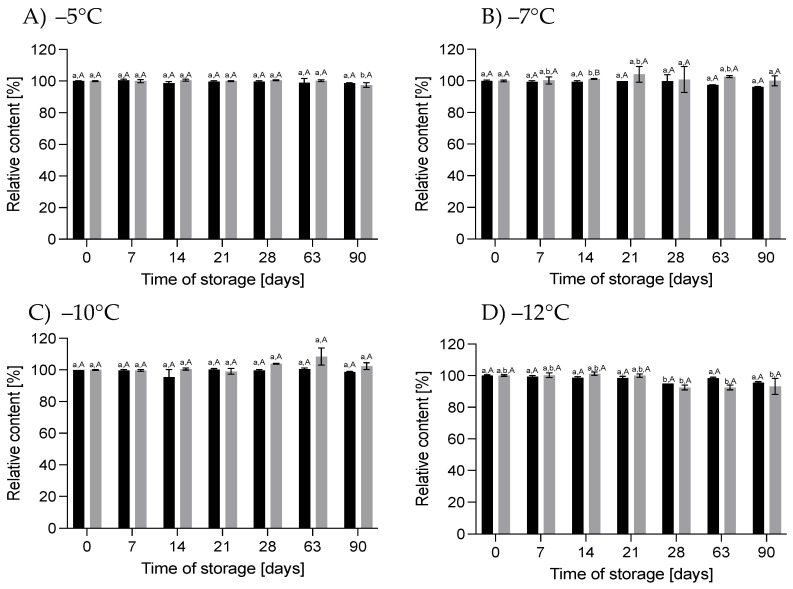
Changes in the relative content of carbohydrates during storage at subzero temperatures under increased pressure (dark bars on each chart) and at atmospheric pressure (light bars on each chart); different lowercase letters (a,b) indicate statistically significant differences among the different storage times for the same treatment, and the capital letters (A,B) indicates statistically significant differences among the various treatments at the same time (*p* < 0.05).

**Figure 6 nutrients-16-01455-f006:**
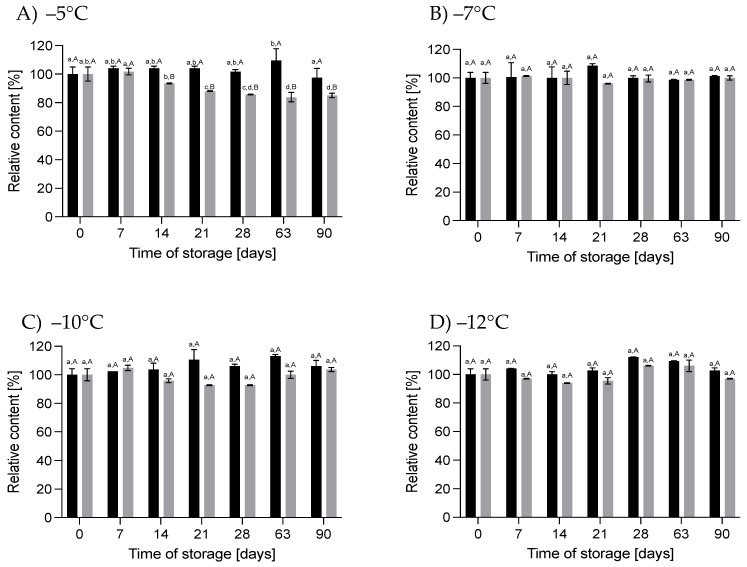
Changes in the relative content of fat during storage at subzero temperatures under increased pressure (dark bars on each chart) and at atmospheric pressure (light bars on each chart); different lowercase letters (a–d) indicate statistically significant differences among the different storage times for the same treatment, and different capital letters (A,B) indicate statistically significant differences among the different treatments at the same time (*p* < 0.05).

**Figure 7 nutrients-16-01455-f007:**
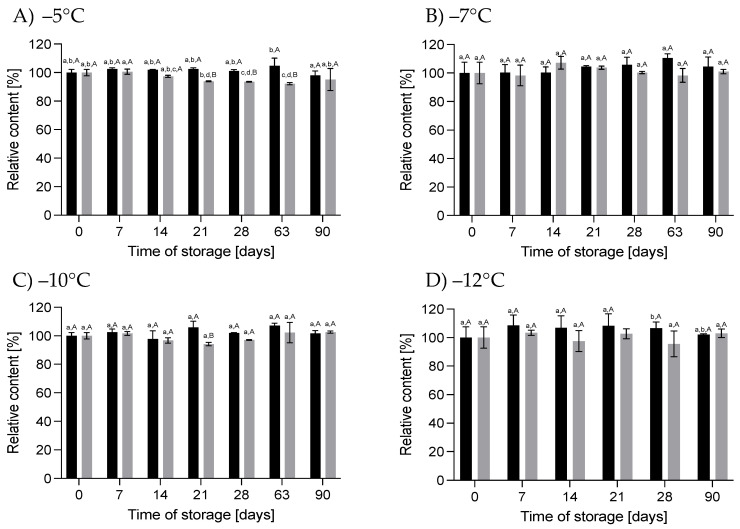
Changes in the relative energy value during storage at subzero temperatures under increased pressure (dark bars on each chart) and at atmospheric pressure (light bars on each chart); different lowercase letters (a–d) indicate statistically significant differences among the different storage times for the same treatment, and different capital letters (A,B) indicate statistically significant differences among the different treatments at the same time (*p* < 0.05).

**Figure 8 nutrients-16-01455-f008:**
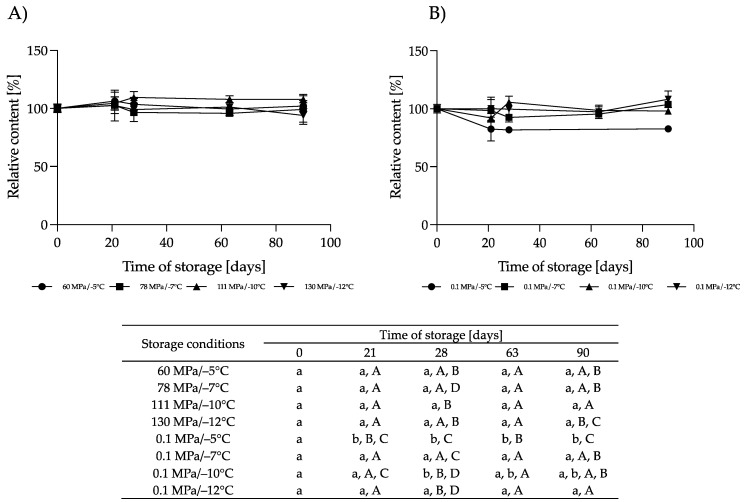
Changes in the relative content of *α-*LAC during storage at subzero temperatures under increased pressure (**A**) and at atmospheric pressure (**B**); different lowercase letters (a,b) indicate statistically significant differences among the different storage times for the same treatment, and different capital letters (A–D) indicate statistically significant differences among the different treatments at the same time (*p* < 0.05).

**Figure 9 nutrients-16-01455-f009:**
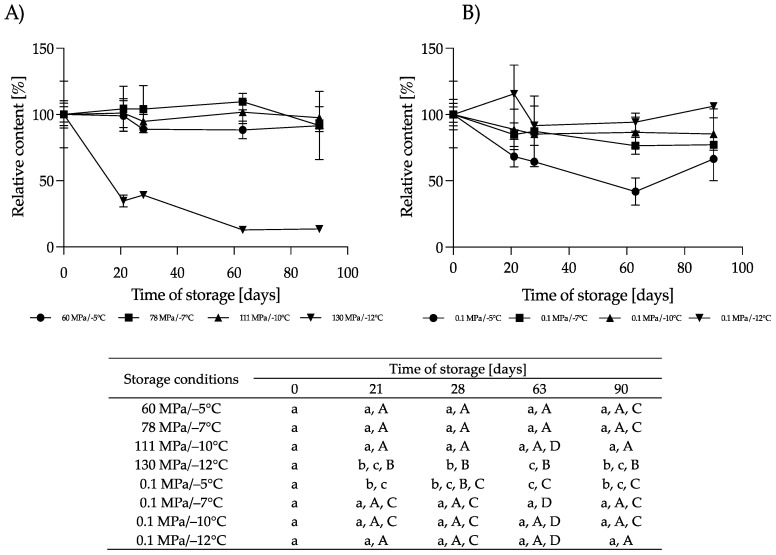
Changes in the relative content of LF during storage at subzero temperatures under increased pressure (**A**) and at atmospheric pressure (**B**); different lowercase letters (a–c) indicate statistically significant differences among the different storage times for the same treatment, and different capital letters (A–D) indicate statistically significant differences among the different treatments at the same time (*p* < 0.05).

**Figure 10 nutrients-16-01455-f010:**
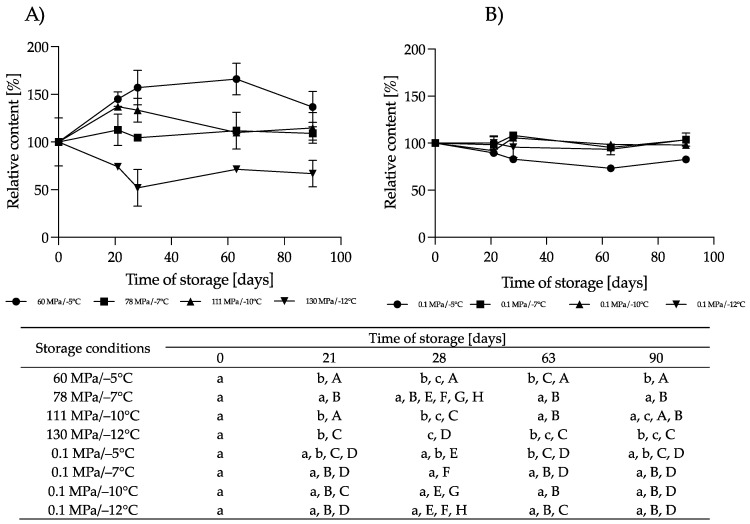
Changes in the relative content of LYZ during storage at subzero temperatures under increased pressure (**A**) and at atmospheric pressure (**B**); different lowercase letters (a–c) indicate statistically significant differences among the different storage times for the same treatment, and different capital letters (A–H) indicate statistically significant differences among the different treatments at the same time (*p* < 0.05).

**Figure 11 nutrients-16-01455-f011:**
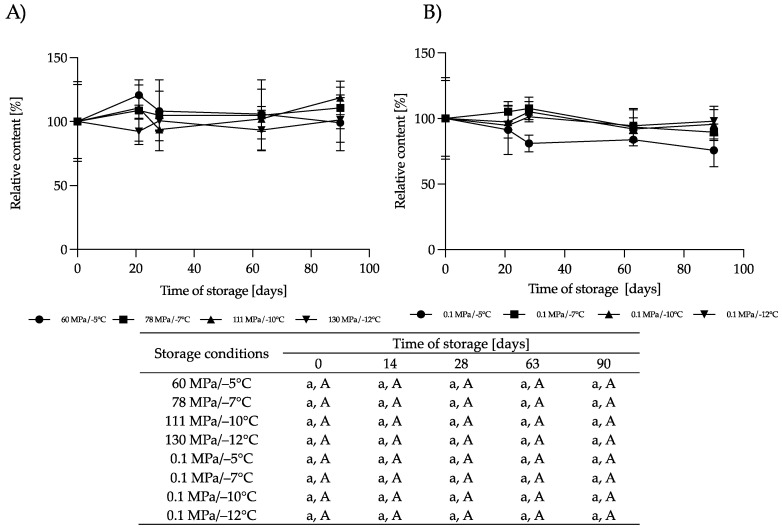
Changes in the relative content of sIgA during storage at subzero temperatures under increased pressure (**A**) and at atmospheric pressure (**B**); the lowercase letter a indicates a lack of statistically significant differences among the different storage times for the same treatment, and the capital letter A indicates a lack of statistically significant differences among the different treatments at the same time (*p* > 0.05).

**Table 1 nutrients-16-01455-t001:** Characteristics of donors participating in the study.

Parameter	Value *
No. of donors	25
Age (years)	30.68 ± 5.14 (19–39)
Gestation number (*n*)	1.64 ± 0.99 (1–4)
Gestation lenght (weeks)	37.3 ± 4.1 (28–41)
Birthweight (g)	3053.4 ± 1093.8 (850–4900)
Apgar scores (Apgar scale)	8.9 ± 1.9 (1–10)
Child’s gender	female 8, male 17
Lactation period (days)	138.2 ± 158.7 (3–803)

* Values presented as arithmetic mean ± SD and range.

**Table 2 nutrients-16-01455-t002:** Macronutrient and bioactive protein content and energy value of initial HM.

Macronutrient	Content
Crude protein [g/100 mL]	1.25 ± 0.155
True protein [g/100 mL]	1.01 ± 0.131
Carbohydrates [g/100 mL]	8.36 ± 0.072
Fat [g/100 mL]	3.96 ± 0.416
Energy value [kcal/100 mL]	75.375 ± 4.256
Bioactive Proteins	
Lactoferrine [g/100 mL]	0.305 ± 0.0147
*α-*lactalbumin [g/100 mL]	0.272 ± 0.0597
Lysozyme [g/100 mL]	0.013 ± 0.0008
sIgA [g/100 mL]	0.107 ± 0.0486

Each value is expressed as mean ± SD of four samples per duplicate (macronutrients) or triplicate (bioactive proteins).

## Data Availability

The data presented in this study are available on a request from the corresponding author due to ethical reason.

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
