# Peer review of "The Effect of Hyperbaric Storage on the Nutritional Value and Retention of Certain Bioactive Proteins in Human Milk"

_nutrients, 2024, doi:10.3390/nu16101455_

Round 1
Reviewer 1 Report
Comments and Suggestions for Authors
The article addresses the important issue of the preserving the composition of human milk during storage. It should be noted that the methodology of the paper does not have a detailed description. There is a lack of information about the involvement of the participants, their consent and the approval of the ethics committee. The principle of participant selection is not really clear. How does the composition of the milk affect the other parameters if one of the participants is lactating for up to 800 days? Some samples are not mature milk samples, which also affects the results. In addition, the stability of the milk proteins, especially the whey proteins, is affected by the preparation of the samples by freezing, thawing and further partial freezing and storage under certain pressures. This certainly complicates the sampling technique and is the answer to such variable results, with increases and decreases in content throughout the storage period. The paper lacks a critical analysis of the data. Even with such a wide range of data, other ways of presenting it should have been sought. The calculation of the arithmetic mean of 2 replicates is not useful.

Author Response
We greatly appreciate the Reviewer’s constructive comments and suggestions that are very helpful for revising and improving our paper, as well as the important guiding significance to our research. We have checked and revised the manuscript carefully according to the comments. Details reply to the comments and suggestions are listed below:
Reviewer 1
The article addresses the important issue of the preserving the composition of human milk during storage.
Comment 1: It should be noted that the methodology of the paper does not have a detailed description. There is a lack of information about the involvement of the participants, their consent and the approval of the ethics committee.
Response: We have completed the methodological data regarding the preparation of milk samples for storage and added information about the involvement of the participants, their consent and the approval of the ethics committee.
Comment 2: The principle of participant selection is not really clear. How does the composition of the milk affect the other parameters if one of the participants is lactating for up to 800 days? Some samples are not mature milk samples, which also affects the results.
Response: As our research was conducted on pooled milk, we did not place restrictions on donors who wished to donate milk for research purposes. The lack of restrictions was also dictated by the fact that we had to collect a large amount of milk at one time in a short period of time. We did not store individual milk samples but pooled them into four large portions so as to obtain a large homogeneous samples, which was further bottled and stored under appropriate pressure-temperature conditions. Under each condition, 2 different batches of milk were stored and taken for testing after 6 storage periods. Thus, a total of about 12 L of milk was required. As each milk sample is very valuable, we decided to use each donated quantity of milk. The portion of milk that came from the 3rd day of lactation accounted for about 3.3% of the total sample, while milk from the 800th day of lactation accounted for about 10%. The results of the analyses (Table 1) indicate that the content of the various components we quantified is within the ranges reported in the literature. Thus, we believe that the combination of milk portions from different lactation periods did not affect the results of our study.
Comment 3: In addition, the stability of the milk proteins, especially the whey proteins, is affected by the preparation of the samples by freezing, thawing and further partial freezing and storage under certain pressures. This certainly complicates the sampling technique and is the answer to such variable results, with increases and decreases in content throughout the storage period.
Response: In our study, we tried to make the conditions of the experiment as close as possible to those that could be used in real conditions. In most HMBs around the world, donors provide milk in frozen form and such milk is stored at -20°C for up to 3 months from the date of expressing. The milk portions are then thawed, combined into large portions, re-portioned and pasteurized. The pasteurized milk is refrozen and stored under these conditions for up to 6 months from the date of expressing. We assumed that if the method we presented were to be applied to HMB, the milk would also be delivered in frozen form and, up to 1 month from the date of expressing, would be thawed, pooled and portioned, and then transferred to hyperbaric storage. Thus, it is impossible to avoid the freezing/thawing steps in the current HMB milk handling regimen as well as in the procedure using hyperbaric storage. In our experiments milk from two batches was stored under each condition. A separate portion was prepared for each measuring point. To clarify these inconsistencies, we have changed the description of sample preparation for storage and added figures.
Comment 4: The paper lacks a critical analysis of the data. Even with such a wide range of data, other ways of presenting it should have been sought. The calculation of the arithmetic mean of 2 replicates is not useful.
Response: As we mentioned above, 2 batches were stored under all conditions. Therefore, the results are the mean value and standard deviations of two separate determinations in duplicate (macronutrients and energy value) or triplicate (bioactive proteins concentration). We added this information to the relevant subsections.
Comment 5: Line 27 - please clarify! lactoferrin, lactoalbumin also belong to nutrients
Response: We changed the keywords. Instead of macronutrients, we used protein, carbohydrates, fats, energy value.
Comment 6: Line 36 - style
Response: Appropriate change has been made to the text of the manuscript.
Comment 7: Line 40 - Style! Previsously mentioned compounds belong to nutrients
Response: We changed the phrase “…of some biologically active components such as LF, LYZ, sIgA and some nutrients.” on “…of some biologically active components like lactoferrine (LF), lysozyme (LYZ), secretory immunoglobuline A (sIgA), other nutrients such as fats or carbohydrates.”
Comment 8: Line 46 and 49 - please clarify
Response: Appropriate changes have been made to the text of the manuscript.
Comment 9: Line 68-71 -
Response: Relevant information has been added to the manuscript.
Comment 10: Line 76 - pooled samples?
Response: No, they were single portions of milk.
Comment 11: Line 77 - Please describe the process, how quick?
Response: Appropriate information has been added to the text.
Comment 12: Line 79 - How many equal samples did you have?
Response: Appropriate information has been added to the text.
Comment 13: Line 101-102 - ?
Response: It is difficult to answer to this comment because we are not entirely sure we understand the comment correctly. This data is available in the manual provided by the device manufacturer.
Comment 14: Line 103 - why?
Response: The manufacturer of the MIRIS device recommends perform duplicate analyses.
Comment 15: Line 135 - In Table 2 - fat
Response: “Lipids” was changed to “fats”.
Comment 16: Line 141: - ?
Response: The sentence “These components in HM have both nutritional and non-nutritional functions, which ensures proper growth and healthy development of infants.” has been change on “These components in HM provide proper nutritional value and are a source of bioactive factors that ensures normal growth and healthy development of infants”.
Comment 17: Line 161 - Do you have substantiation for protein amount reduction? How homogeneous is the product as sampled?
Response: We searched the literature for an explanation of why there is a reduction in protein content after the HoP process. Unfortunately, in none of the papers do the authors substantiate why this happens.
Comment 18: Figure 1 - -10C
Response: Changed 10°C to -10°C
Comment 19: Line 181 - and WP denaturation?
Response: In the paper we cited, we did not find information that denaturation of whey proteins occurs under these conditions.
Comment 20: Line 204 - what exactly?
Response: The sentence “…. and we noted that in these conditions some changes in the concentration of carbohydrates took place.“ was changed on “…. and we noted that in these conditions decreasing in the concentration of carbohydrates took place.”
Comment 21: Line 205 - and microorganisms and their growing?
Response: It is difficult for us to address this comment due to the fact that no microbial growth is observed when milk is stored at subzero temperatures under both atmospheric and moderate pressure. Our results show that storage in subzero temperatures under atmospheric pressure has little effect on the numbers of milk microbiota (their growth is inhibited) or some populations (for example, coagulase-positive staphylococci) are reduced. The results regarding survival of HM microbiota under HT-ST are just under evaluation in another journal.
Comment 22: Line 237-239 - Please reconsider. You have many treatments: freezing, thawing, and HS-ST. How these treatments affected the stability of the compositions and the sampling practices.
Response: The fat content results are presented as a relative value to the control (non-stored) samples, so the freezing/thawing process that took place in the first stage of sample preparation for testing (Fig. 1) was unlikely to affect the final result.
Comment 23: Figure 4 - Do you have substantiation for the decreasing and increasing the amount of analyzed compounds in one sample during storage?
Response: The results presented in manuscript are the average of 4 measurements (two independent milk samples analyzed in duplicate), which probably explains the fluctuations of the analyzed compounds. However, statistical analysis showed that these differences are usually not significant.
Comment 24: Line 248 - Line 218 - no significant
Response: We rechecked the results of the statistical analysis and noticed that in the case of Figure 5A, incorrect labels were entered. We are very sorry for this mistake. We have once again checked the correctness of all statistical determinations on all figures in the manuscript.
Comment 25: Line 251 - Even within a single treatment during storage, your results show a large variation. This is probably due to the samples having freeze down. Also, the description of the freezing of the sample is not sufficient. Were you able to make sure that the temperature was kept at a constant level during storage?
Response: We have completed the information in the Materials and methods section. Temperature was monitored throughout the storage period using an electronic sensor integrated into the storage chamber.
Comment 26: Line 258 - Did you measure bioactivity!
Response: No, we measured the content of bioactive proteins. Appropriate change has been made in the title of this subsection.
Comment 27: Line 264 - human milk? Frozen state?
Response: Following the suggestion of the second reviewer, Tables 2 and 3 have been combined and the title has been changed.
Comment 28: Table 3 - It is recommended that you find a more appropriate way of displaying the data as you have such a high SD. Please note that you only have 2 replicates. How stable is your data?
Response: We are very sorry, but during the preparation of the manuscript we changed the unit in which we reported the bioactive protein concentration and unfortunate forgot to do the same with the standard deviation. These data are the mean value and standard deviation obtained from 4 different control samples, each analyzed in triplicate. In addition, the mean values obtained, as well as the individual results (separately for each control sample), are consistent with literature data.
Comment 29: Line 273 – Why?
Response: This is our mistake. As the statistical analysis indicates, this change was not significant.
Comment 30: Line 298-230 - please explain it
Response: Such a reduction in LF content is likely due to denaturation of this protein under these conditions. A corresponding explanation has been added to the text of the manuscript.
Comment 31: Figure 7 - Have you explanation of the results after 60 days and 90 days of storage 0.1 MPa -5C?
Response: Unfortunately, we could not find in the literature a reliable explanation for the differences in LF content after 60 and 90 days of storage at 0.1 MPa/-5°C. Lactoferrin is secreted into milk by epithelial cells of mammary gland tissue and by activated leukocytes (Riskin et al. 2012 https://doi.org/10.1038/pr.2011.34). Perhaps, after 90 days of storage, there was significant damage to leukocytes (due to the formation of large ice I crystals under these conditions) from which LF was released, so that there was a noticeable increase in the content of this protein.
Comment 32: Line 353 - Lack of explanation.
Response: We have added possible explanations for the different levels of sIgA inactivation after the pasteurization proces.
Comment 33: Line 362 - ? are you sure?
Response: Increasing the pressure to 207 MPa shifts the crystallization point of water to about -22°C. This was described in the early 20th century by Bridgman 1912 and later by Hobbs 1974 (Ice Physics, Clarendon Press, Oxford) and Rubinsky et al., 2005 (https://doi.org/10.1016/j.cryobiol.2004.12.002).
Comment 34: Line 367 - and size of ice crystals?
Response: We added information that in these conditions large ice crystals are formed.

Reviewer 2 Report
Comments and Suggestions for Authors This manuscript carried out a study that replace tradational pasteurization of human milk by high pressure processing (HPP), and determined the changes in the content of macronutrients and selected bioactive proteins. This study provides data that is of value for human milk-related study.The main issue of this study, as the authors already hinted, is that the effects of the proposed methods on biohazards such as viruses and microbiota are not evaluated. The validity should be first proven, then the nutritional value evaluated afterwards.
Other issue lies on that the control group is not clearly expressed. Whether the traditional method, HoP, used as control? Figure 2-5 did not indicate HoP as a valid control group. Why use atmospheric pressure without HoP as control?
From the raw data expressed in the manuscript I did not see significant differences in these analyses. If there are significant differences between any of these groups, you should highlight with specific nutrients, and conditions, in conclusions.
Line 105: is Kinetex XB-C18 chromatographic column used in analysis of 𝛼-LAC, LF and LYZ content? no sample treatment is introduced here, so I assume the intact protein is analyzed instead of proteomics. it is unlikely to use a C18 column instead of a C4 column for such large molecules.
Minor: The entire manuscript needs language improvement and improve style. Some of the issues were:
Line 105 and alike: no capitalization needed for "High Performance Liquid Chromatography"
Line 63: LF, LYZ, 𝛼-LAC and sIgA, need complete spelling for them. Abstract and main text is separate part and need to define abbreviations separately.
Line 63: What are these proteins? Why they impact HM instead of other proteins?
Line 83: Patent Application Number P.447629, please give this as a reference.
Line 95: give reference for Miris Human Milk Analyzer User Manual
Line 92 and 99: is MIRIS being as Miris
Figure 1 can be not starting from zero for Y-axis to show a clear difference.
Line 360: Subtitle with no period
Figure 6-9: These figures can be combined together. For the Tables part, it should also combined into a new table. Comments on the Quality of English Language
I think extensive editing of English language required, because I found many scientific style errors.
Author Response
We greatly appreciate the Reviewer’s constructive comments and suggestions that are very helpful for revising and improving our paper, as well as the important guiding significance to our research. We have checked and revised the manuscript carefully according to the comments. Details reply to the comments and suggestions are listed below:
Reviewer 2
This manuscript carried out a study that replace tradational pasteurization of human milk by high pressure processing (HPP), and determined the changes in the content of macronutrients and selected bioactive proteins. This study provides data that is of value for human milk-related study.
Comment 1: The main issue of this study, as the authors already hinted, is that the effects of the proposed methods on biohazards such as viruses and microbiota are not evaluated. The validity should be first proven, then the nutritional value evaluated afterwards.
Response: We mentioned that under these conditions there is a complete elimination of the milk microbiota. We presented the results of the microbiological determinations at the conference ([9] Malinowska-Pańczyk, E.; Mazur, K.; Martysiak-Żurowska, D. A New Method of Human Milk Preservation: Storage in Unfrozen State under High Pressure-Subzero Temperature Conditions. In Proceedings of the A new method of human milk preservation: storage in unfrozen state under high pressure-subzero temperature conditions; Italy, 2023.) and they are currently under evaluation in another journal.
Comment 2: Other issue lies on that the control group is not clearly expressed. Whether the traditional method, HoP, used as control? Figure 2-5 did not indicate HoP as a valid control group. Why use atmospheric pressure without HoP as control?
Response: Data on changes in the composition of human milk after the HoP have been presented in many publications, and most of these results confirm the negative effects of heating on HM. Due to the fact that for our experiments we needed to accumulate almost 12 L of milk in a short period of time (less than 1 month) we decided not to conduct an experiment in which we would study the changes that occur in milk stored at -20°C after the pasteurization process. Storing the samples at subzero temperatures (-5, -7, -10, -12°C) under atmospheric pressure was carried out to see if the changes that occur under moderate pressure and subzero temperatures are related only to the effect of pressure.
Comment 3: From the raw data expressed in the manuscript I did not see significant differences in these analyses. If there are significant differences between any of these groups, you should highlight with specific nutrients, and conditions, in conclusions.
Response: We rewritten the conclusions to highlight that the nutritional value of the milk did not change during storage
Comment 4: Line 105: is Kinetex XB-C18 chromatographic column used in analysis of ?-LAC, LF and LYZ content? no sample treatment is introduced here, so I assume the intact protein is analyzed instead of proteomics. it is unlikely to use a C18 column instead of a C4 column for such large molecules.
Response: Preparation of milk samples for this type of analysis mainly involves lipids separation by centrifugation. This approach has so far been used by several authors studying human milk (Jackson et al., 2004 https://doi.org/10.1016/j.jnutbio.2003.10.009; Chowanadisai et al., 2005 https://doi.org/10.1016/j.jnutbio.2004.12.010; Santos and Ferreira 2007 https://doi.org/10.1016/j.ab.2006.12.002). Additionally, the samples were not filtered before analysis because, as in other studies, it was found that lactalbumin may adhere to the filter (Jackson et al., 2004). Some types of columns packing such as C3 (propyl), C4 (butyl), C8 (octyl), C18 (octadecyl), cyanopropyl and phenyl groups are available for determination of proteins (Wu and Greenblatt, 1995 https://doi.org/10.1016/0021-9673(94)01139-6). Maybe the C4 stationary phases are more popular, but the C18 phases are also used. Xiao-yu et al. (2012) (https://doi.org/10.1016/S1006-8104(13)60026-4) compared these two types of columns on the basis of bovine milk analysis (Fig. 1). According to their results, both columns are suitable for this type of analysis.
(Figure copied from Xiao-yu et al. 2012)
Wu and Greenblatt (1995) also studied the effect of different stationary phases on protein separation and they similarly demonstrated the suitability of the C18 column (Fig. 5).
(Figure copied from Wu and Greenblatt, 1995)
In the presented research, we used the Kinetex XB-C18 column because it enabled the separation and quantification of target analytes. The chromatograms of an example milk sample and samples of original standards obtained during the tests are presented below.
Minor: The entire manuscript needs language improvement and improve style. Some of the issues were:
Comment 5: Line 105 and alike: no capitalization needed for "High Performance Liquid Chromatography"
Response: Appropriate changes have been made according to Reviewer suggestion.
Comment 6: Line 63: LF, LYZ, ?-LAC and sIgA, need complete spelling for them. Abstract and main text is separate part and need to define abbreviations separately.
Response: Appropriate changes have been made according to Reviewer suggestion.
Comment 7: Line 63: What are these proteins? Why they impact HM instead of other proteins?
Response: To investigate changes in HM during storage under different conditions, we chose bioactive proteins whose percentages, relative to other bioactive proteins present in milk, remain high. In addition, they are particularly important in supporting the immature immune system of newborns and infants.
Comment 8: Line 83: Patent Application Number P.447629, please give this as a reference.
Response: Appropriate change has been made according to Reviewer suggestion.
Comment 9: Line 95: give reference for Miris Human Milk Analyzer User Manual
Response: Reference was added to the text.
Comment 10: Line 92 and 99: is MIRIS being as Miris
Response: Changed MIRIS to Miris as the Miris website doesn’t use capital letters in its name.
Comment 11: Figure 1 can be not starting from zero for Y-axis to show a clear difference.
Response: We left the figure unchanged due to the fact that in the other figures the Y axis starts from 0.
Comment 12: Line 360: Subtitle with no period
Response: Period has been removed.
Comment 13: Figure 6-9: These figures can be combined together.
Response: Merging Figures 6-9 would cause them to have to be significantly reduced in size, making them less readable. Therefore, we left the Figures unchanged.
Comment 14: For the Tables part, it should also combined into a new table.
Response: The tables have been merged.

Round 2
Reviewer 2 Report
Comments and Suggestions for Authors
Most of my previous comments were responded properly. However there are still minor style issue remains, it should be corrected during the rest of the process.
some of the issues I found were:
Reference style need to be improved:
Miris Human Milk Analyzer User Manual is a website address, should also list as a formal reference with the style according to your journal. Reference of Patent Application Number P.447629, missing key information such as country, date, etc. Figure 8 and alike: the tables still attached to the Figure. I am not sure if it should be separated, at least it should be presented with no red underline.
Author Response
We would like to thank the Reviewer of our manuscript for very constructive remarks on the original version of the paper.
Most of my previous comments were responded properly. However there are still minor style issue remains, it should be corrected during the rest of the process.
Some of the issues I found were:
Comment 1: Reference style need to be improved
Response: References have been carefully checked and corrected according to the journal's requirements.
Comment 2: Miris Human Milk Analyzer User Manual is a website address, should also list as a formal reference with the style according to your journal.
Response: The link to the website on which the Miris Human Milk Analyzer User Manual is located has been added to the References section.
Comment 3: Reference of Patent Application Number P.447629, missing key information such as country, date, etc.
Response: The relevant patent application information has been completed.
Comment 4: Figure 8 and alike: the tables still attached to the Figure. I am not sure if it should be separated, at least it should be presented with no red underline.
Response: We decided to present the results of the statistical analysis in the form of tables under the graphs for the better readability. The red lines have been removed.